# PPP3CB overexpression mediates EGFR TKI resistance in lung tumors via calcineurin/MEK/ERK signaling

Sylvie Gazzeri[1] , Nadiia Zubchuk[1] , Elodie Montaudon[2], Fariba Nemati[2], Sarah Huot-Marchand[1], Giulia Berardi[3], Amelie Pucciarelli[1], Yassir Dib[1], Dylan Nerini[1], Christiane Oddou[9], Mylène Pezet[10] , Laurence David-Boudet[4], Camille Ardin[1,3], Florence de Fraipont[1,5] , Antonio Maraver[6], Nicolas Girard[7], Didier Decaudin[2,8], Anne-Claire Toffart[1,3], Beatrice Eymin[1]

**Despite initial high response rates to first-line EGFR TKI, all non–small-cell lung cancer (NSCLC) with EGFR-activating mutation will ultimately develop resistance to treatment. Identification of resistance mechanisms is critical to adapt treatment and improve patient outcomes. Here, we show that a *PPP3CB* transcript that encodes full-length catalytic subunit 2B of calcineurin accumulates in EGFR-mutant NSCLC cells with acquired resistance against different EGFR TKIs and in post-progression biopsies of NSCLC patients treated with EGFR TKIs. Neutralization of *PPP3CB* by siRNA or inactivation of calcineurin by cyclosporin A induces apoptosis in resistant cells treated with EGFR TKIs. Mechanistically, EGFR TKIs increase the cytosolic level of calcium and trigger activation of a calcineurin/MEK/ERK pathway that prevents apoptosis. Combining EGFR, calcineurin, and MEK inhibitors overcomes resistance to EGFR TKI in both in vitro and in vivo models. Our results identify PPP3CB overexpression as a new mechanism of acquired resistance to EGFR TKIs, and provide a promising therapeutic approach for NSCLC patients that progress under TKI treatment.**

## Introduction

EGFR-targeted therapy based on EGFR TKI has demonstrated unprecedented results on the survival of non–small-cell lung cancer (NSCLC) patients with activating EGFR mutations. However, even in the 60–80% of responders, acquired resistance to treatment occurs. In a large proportion of patients treated with

first- and second-generation TKIs, resistance is linked to the acquisition of a secondary EGFR T790M mutation (40–60%) (Sequist et al, 2011). Third-generation EGFR TKIs that selectively target EGFR mutations including the T790M have been developed. The representative one, osimertinib (AZD9291 or Tagrisso), has demonstrated robust clinical activity as a second-line treatment in NSCLC patients with EGFR T790M mutation (Mok et al, 2017). It is now also the standard of care as a first-line treatment for all patients with locally advanced or metastatic EGFR-mutant NSCLC regardless of the T790M mutation status (Soria et al, 2018; Ramalingam et al, 2020). Unfortunately, similar to previous-generation EGFR TKIs, acquired resistance to osimertinib eventually develops and therefore limits its long-term effectiveness. Although not fully understood, particularly after first-line osimertinib treatment, those resistance mechanisms are complex and multifaceted, and include EGFR-dependent and EGFR-independent mechanisms (Passaro et al, 2021; Blaquier et al, 2023).

Calcineurin is a calcium-dependent serine/threonine phosphatase that links changes in intracellular calcium levels to protein phosphorylation states and signaling processes. Activation of calcineurin induces the dephosphorylation of a variety of substrates that include transcription factors, proteins involved in cell cycle and apoptosis, cytoskeletal proteins, scaffolding proteins, and channel membrane proteins and receptors (Li et al, 2011). Previous studies have shown that calcineurin is frequently activated in various cancers and induces signaling pathways that stimulate cell proliferation and migration, as well as metastasis (Jauliac et al, 2002; Buchholz et al, 2006; Minami et al, 2013; Tie et al, 2013; Quang et al, 2015; Manda et al, 2016; Shoshan et al, 2016). In agreement with this pro-tumorigenic

---

[1]University Grenoble Alpes, Inserm U1209, CNRS UMR 5309, Team RNA Splicing, Cell Signaling and Response to Therapy, Institute for Advanced Biosciences, Grenoble, France   [2]Laboratory of Preclinical Investigation, Translational Research Department, Institut Curie, PSL Research University, Paris, France   [3]Department of Pneumology and Physiology, Grenoble-Alpes University Hospital, Grenoble, France   [4]Department of Cytology and Pathology, Grenoble-Alpes University Hospital, Grenoble, France   [5]Medical Unit of Molecular Genetic (Hereditary Diseases and Oncology), Grenoble-Alpes University Hospital, Grenoble, France   [6]Institut de Recherche en Cancérologie de Montpellier, INSERM U1194-ICM-Université de Montpellier, Montpellier, France   [7]Institut du Thorax Curie-Montsouris, Institut Curie, Paris, France   [8]Department of Medical Oncology, Institut Curie, Paris, France   [9]University Grenoble Alpes, Inserm U1209, CNRS UMR 5309, Team Epigenetics, Immunity, Metabolism, Cell Signaling and Cancer, Institute for Advanced Biosciences, Grenoble, France   [10]University Grenoble Alpes, Inserm U1209, CNRS UMR 5309, Platform MicroCell, Institute for Advanced Biosciences, Grenoble, France

Correspondence: sylvie.gazzeri@univ-grenoble-alpes.fr; beatrice.eymin@univ-grenoble-alpes.fr

role, some studies have reported that calcineurin inhibitors could be potential anticancer agents (Medyouf et al, 2007; Siamakpour-Reihani et al, 2011; Kawahara et al, 2015a, 2015b). Calcineurin (CaN) is a heterodimer that consists of a catalytic subunit calcineurin A and a regulatory subunit calcineurin B (Rusnak & Mertz, 2000; Creamer, 2020). Three isoforms of calcineurin A have been described, namely, CaNα, CaNβ, and CaNγ. They are encoded by three distinct serine/threonine-protein phosphatase 2B catalytic subunit (PPP3C) genes (*PPP3CA*, *PPP3CB*, and *PPP3CC*, respectively), *PPP3CB* being the predominant form expressed in most non-neuronal cell types.

Our data identify PPP3CB overexpression as a new mechanism of acquired resistance to EGFR TKI in EGFR-mutant NSCLC, and provide a promising therapeutic approach to overcome acquired resistance.

# Results

## PPP3CB Ex16 splice variant accumulates in EGFR TKI–resistant NSCLC cells and prevents apoptosis in response to EGFR TKI

Looking for abnormal RNA splicing events as novel alterations during acquisition of resistance on EGFR TKI treatment, we previously performed high-throughput RNA sequencing in 3 gefitinib- and 3 dacomitinib-resistant PC9-derived clones (PC9/GR1 to GR3 and PC9/DR1 to DR3, respectively) and showed altered splicing profiles as compared to sensitive parental cells (Hatat et al, 2022). PC9 cells are highly sensitive to EGFR TKI and are frequently used as a model for studying the resistance of NSCLC to EGFR TKI (Della Corte et al, 2018; Taniguchi et al, 2019; Vojnic et al, 2019; Lin et al, 2023). Among exon skipping events that were validated by RT–PCR, 11 were common to cells with acquired resistance to gefitinib or dacomitinib. To study whether these splicing changes also occur after treatment with third-generation EGFR TKI, we performed RT–PCR experiments in 3 PC9-derived clones with acquired resistance to osimertinib (PC9/OR1 to OR3). Eight of the 11 exon skipping events were also found in the PC9/OR clones compared with parental PC9 cells (Fig S1), including *PPP3CB* (Fig 1A). *PPP3CB* encodes the catalytic subunit of calcineurin, a pro-tumorigenic protein whose pharmacological targeting has been reported to have potent antitumor effects in various cancers (Medyouf et al, 2007; Siamakpour-Reihani et al, 2011; Kawahara et al, 2015a, 2015b). We observed the higher expression of a *PPP3CB* splice variant that retains exon 16 (further called *PPP3CB* Ex16) in 8 of 9 resistant clones as compared to parental PC9 cells (Fig 1A). Further study was performed on resistant PC9/GR1, PC9/DR1, and PC9/OR3 cells (subsequently named PC9/GR, PC9/DR, and PC9/OR) that exhibited the higher exon 16 inclusion rate based on the RT–PCR data. Several splice variants of *PPP3CB* have been reported. According to Ensembl, only one splice variant retains exon 16 (PPP3CB-202 transcript) and encodes the canonical full-length isoform of PPP3CB. Consistent with RT–PCR data, PPP3CB protein accumulation was confirmed in resistant cells as compared to parental sensitive PC9 cells (Fig 1B).

To assess whether and how up-regulation of *PPP3CB* Ex16 is involved in resistance to EGFR TKI, we transfected PC9/GR, PC9/DR, and PC9/OR cells with a siRNA that targets the exon 16 (siEx16) or with a siRNA that targets the exon 15/17 junction (siEx15/17, to inhibit RNAs that do not contain exon 16) or with a control siRNA (siCtl). Of note, as exon 16 is only 30 base pairs, only one siRNA against exon 16 could be designed. Cells were then incubated with or without EGFR TKI, and apoptosis was studied by active caspase 3 staining followed by flow cytometry to detect and quantify apoptotic cells. In all resistant cell lines, transfection with siEx16 greatly enhanced apoptotic levels in response to EGFR TKI as compared to siCtl-transfected cells (Fig 1C). In contrast, transfection with siEx15/17 did not or minimally enhance active caspase 3–positive cells after EGFR TKI treatment. Using MTS assays, we also showed that *PPP3CB* Ex16 inhibition using siEx16 strongly reduced cell viability in response to EGFR TKIs (Fig 1D). To further investigate the role of *PPP3CB* Ex16 accumulation in mediating EGFR TKI resistance, PC9 cells transduced with retroviral vectors driving the expression of either full-length PPP3CB-tagged green fluorescent protein (GFP) or GFP as a control were generated (Fig 1E, left panel). The quantification of active caspase 3–positive cells by FACS showed that apoptosis levels in response to EGFR TKIs were significantly reduced in PPP3CB-GFP cells as compared to GFP control cells (Fig 1E, right panel). In addition, clonogenic assays revealed that PC9-PPP3CB-GFP cells were more resistant to osimertinib-induced cell growth inhibition than PC9-GFP cells (Fig 1F). Together, these results indicated that up-regulation of the *PPP3CB* Ex16 splice variant protects PC9-sensitive cells from the pro-apoptotic effects of various EGFR TKIs.

## Expression of the *PPP3CB* Ex16 transcript in human biopsies from NSCLC patients treated with EGFR TKI

To study whether *PPP3CB* Ex16 accumulation occurs at disease progression under EGFR TKI treatment, we took advantage of a retrospective cohort of 43 EGFR-mutant lung adenocarcinoma patients with paired tissue sections (pre- and post-progression after EGFR TKI treatment). As PPP3CB Western blots revealed the presence of several bands that prevented the specific detection of *PPP3CB* Ex16 using IHC, we decided to develop an RNA-based technology. RNA BaseScope is an in situ hybridization assay for the detection of RNA in the formalin-fixed paraffin-embedded (FFPE) tissue. In this assay, an RNA probe was designed to specifically target exon 16 of *PPP3CB* (Fig 2A). First, we transfected PC9/OR cells with control (siCtl) or siEx16 siRNA and analyzed *PPP3CB* mRNA signals (pink dots) in FFPE cells. The results showed a strong decrease in the percentage of cells with dots in PC9/OR *PPP3CB* Ex16-knockout cells compared to cells transfected with control siRNA (Fig 2B and C) demonstrating the specificity of our probe. Then, we analyzed the expression of *PPP3CB* mRNA in human biopsies (Table 1). Most paired samples were from patients treated with only one-line EGFR TKI including three cases with osimertinib (n = 25). Others were from patients who also received a second EGFR TKI (n = 5) and/or chemotherapy (n = 13) during the course of their treatment. The rebiopsy was performed when patients experienced tumor progression on treatment.

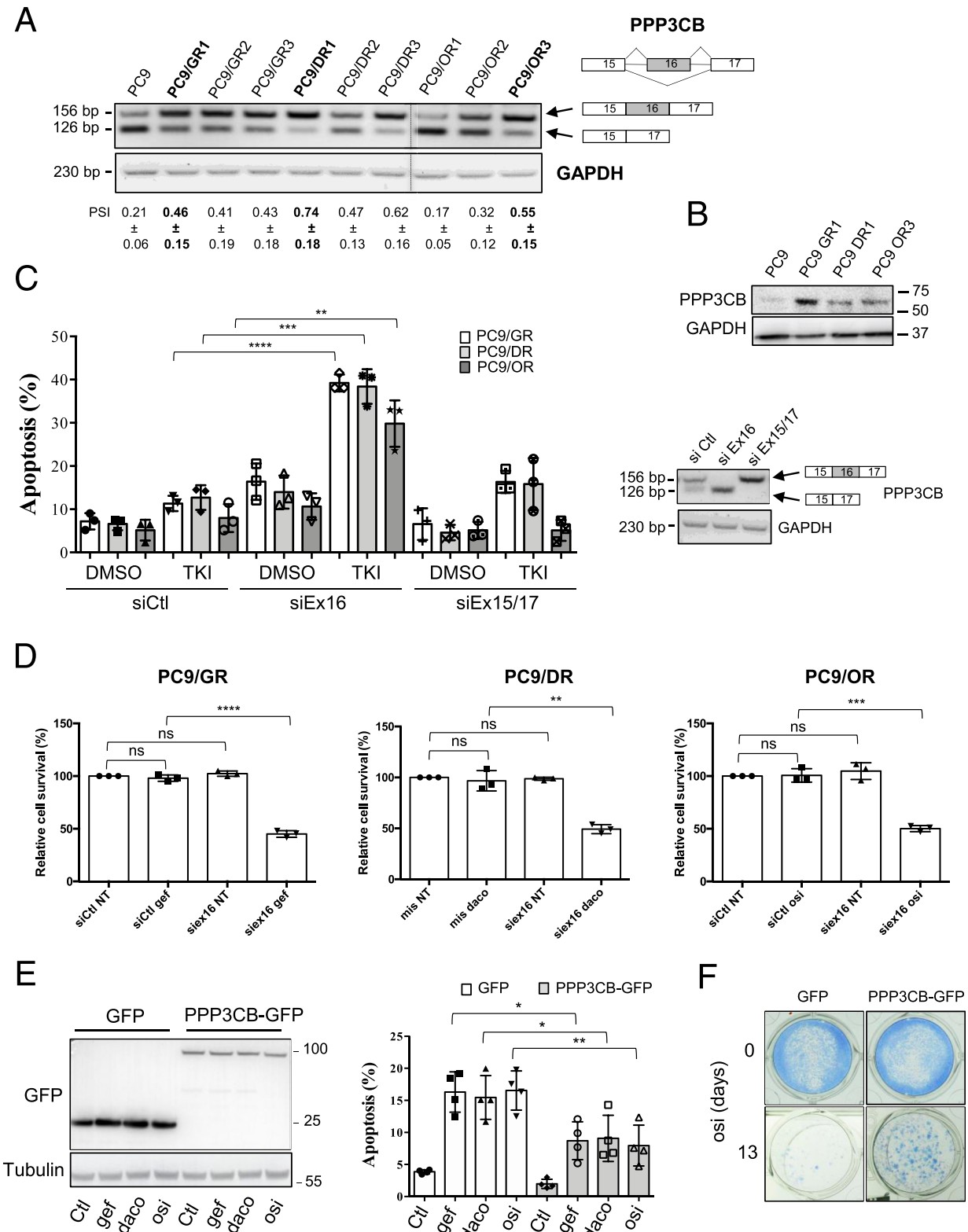

**Figure 1. Cells with acquired resistance to EGFR TKIs accumulate a *PPP3CB* splice variant isoform that retains exon 16.**
**(A)** RT–PCR showing accumulation of PPP3CB-Ex16 in clones with acquired resistance to EGFR TKI (PC9/GR, PC9/DR, PC9/OR) as compared to parental sensitive cells (PC9). Percent spliced-in (PSI) is indicated below the PCR blot. Data are presented as the mean ± SD. n = 3 biological replicates. The dotted line highlights cropped images from the same gel. **(B)** Western blot showing accumulation of PPP3CB protein in resistant cells (PC9/GR1, PC9/DR1, PC9/OR3) compared with sensitive cells (PC9). GAPDH was used as a loading control. **(C)** Resistant cells were transfected with a siRNA targeting exon 16 (siEx16) or the exon 15/17 junction (siEx15/17) of *PPP3CB* or with a

Considering paired samples in the full cohort, 26% of patients (11/43) exhibited an increased percentage of cells with PPP3CB dots in post-treatment samples compared with pre-treatment samples (Table 1 and Fig 2D and E). This increase was not found when patients received chemotherapy. It reached 40% (10/25) in the subcohort of patients treated with a single line of EGFR TKI, and in 20% (5/25) of cases, other concurrent resistance mechanisms including EGFR T790M mutation and MET amplification were not observed (Fig 2F). Together, these results demonstrated that PPP3CB Ex16 mRNA accumulates in tumors that progress under EGFR TKI treatment. The first-line EGFR TKI treatment duration was not different according to the PPP3CB expression level in post-treatment samples (Fig 2F). In pre-treatment samples, the basal level of PPP3CB Ex16 mRNA was heterogeneous (Table S1). Although there was no statistical difference in the duration of first-line EGFR TKI treatment when the PPP3CB basal level was considered, we noticed that patients with a low PPP3CB basal level had numerically longer treatment duration (median = 599 mo with a low PPP3CB level versus 277 mo with a high PPP3CB level) (Table 2).

### Cyclosporin A restores the antitumor effects of EGFR TKIs by promoting apoptosis in PC9-resistant clones

To further explore the role of PPP3CB in resistance to EGFR TKIs, we used cyclosporin A (CsA), a specific calcineurin inhibitor. To evaluate its positive effect in association with EGFR TKI, the concentration of CsA was defined for each type of experimentation in order to have a minimal effect on the inhibition of cell growth when used as monotherapy. Viability assays were performed in PC9/GR, PC9/DR, and PC9/OR cells cultured with an increasing concentration of EGFR TKIs in the presence or absence of CsA. In all resistant cell lines, the EGFR TKI and CsA combinations strongly reduced cell viability compared with EGFR TKIs alone (Fig 3A, upper panel). Of note, a moderate cytotoxicity of CsA alone was observed with the concentrations used (Fig 3A, lower panel). To go further, we looked at the effects of the EGFR TKI/CsA combinations on apoptotic levels using active caspase 3 staining and FACS analysis. Co-treatment with CsA and EGFR TKIs strongly enhanced apoptosis, whereas each single drug had a minimal effect (Fig 3B). To confirm the growth inhibitory effects of the EGFR TKI/CsA combination, colony formation assays were performed. Although the resistant cell lines did not grow as well-identified colonies, the EGFR TKI and CsA co-treatment was highly effective in decreasing cell growth, whereas each single drug had limited effects in the range of the concentrations used

(Fig 3C). The same results were obtained using 3D cell culture experiments (Fig 3D). Taken together, these results showed that pharmacological inhibition of calcineurin by cyclosporin A restores the antitumor effects of EGFR TKI by inducing apoptosis in PC9-resistant clones.

### EGFR TKI activates a $Ca^{2+}$/calcineurin/MEK/ERK1/2 pathway in PC9-resistant clones

As previously stated, resistance mechanisms to EGFR TKI are complex and multifaceted. Importantly, resistant PC9/GR, PC9/DR, and PC9/OR cells did not carry known resistance mechanisms including EGFR mutations (T790M, C797S), MET/HER2 amplifications, or Ras/Raf/PIK3CA mutations/rearrangements (see the Materials and Methods section). So, we first analyzed EGFR-dependent signaling cascades including ERK1/2, AKT, and STAT3. In both sensitive and resistant cells, pAKT and pSTAT3 were similarly regulated by EGFR TKIs. In contrast, although pERK levels were strongly diminished by EGFR TKIs in PC9-sensitive cells as expected, they were increased or maintained in all resistant PC9/GR, PC9/DR, and PC9/OR cells despite EGFR inhibition (Figs 4A and S2A and B). These results indicated that ERK remains activated in EGFR TKI–treated resistant cells. Of note, a previous study reported that STAT3 activation was involved in acquired resistance to gefitinib in EGFR-mutant NSCLC cells (Shou et al, 2016). As we did not observe variation in basal pSTAT3 levels between sensitive and resistant cells, as well as after EGFR TKI treatment, we did not investigate further this signaling pathway. In agreement with a pro-tumorigenic role of the MEK/ERK signaling pathway in our resistant cells, genetic (siRNA) or pharmacological (trametinib) MEK/ERK inhibition induced apoptosis on EGFR TKI treatment as evidenced by PARP and caspase 3 cleavages (Figs 4B–D and S3A). Colony formation assays (Fig 4E) and 3D cell culture experiments (Fig 4F) confirmed the antitumor effect of the trametinib/EGFR TKI combinations. Therefore, we investigated the role of calcineurin in MEK/ERK activation. We showed that EGFR TKIs reduced p-ERK1/2 levels in resistant cells transfected with siEx16 but not with siEx15/17 (Figs 4G and S3B). In addition, co-treatment with EGFR TKIs and CsA strongly diminished p-ERK1/2 levels compared with each single drug, whereas CsA had no significant effect on phosphorylation of EGFR (Figs 4H and S3C). On the contrary, the enforced expression of PPP3CB-GFP in PC9-sensitive cells prevented or slowed down the decrease of p-ERK1/2 levels upon EGFR TKIs as compared to control GFP-expressing cells (Figs 4I and S3D). Together, these results supported a scenario whereby the increased expression of calcineurin in PC9-resistant

control siRNA (siCtl), and treated with appropriate EGFR TKI (either gefitinib, dacomitinib, or osimertinib, 0.1 μM) for 72 h. Quantification of apoptotic rates was determined by active caspase 3 staining and flow cytometry (left panel). Data are presented as the mean ± SD, n = 3 biological replicates. ****P < 0.0001, ***P < 0.001, **P < 0.01, unpaired t test. Neutralization of PPP3CB isoforms was studied by RT–PCR in PC9/OR (right panel). GAPDH was used as a control. **(D)** Neutralization of PPP3CB Ex-16 (siEx16) inhibits cell viability (MTS assay) in response to EGFR TKIs (0.1 μM, 72 h). Data are presented as the mean ± SD, n = 3 biological replicates. ****P < 0.0001, ***P < 0.001, **P < 0.01, ns, not significant, unpaired t test. **(E)** PC9/GFP and PC9/PPP3CB-GFP cells were cultured for 36 h with gefitinib (0.1 μM), dacomitinib (0.05 μM), or osimertinib (0.1 μM). Western blotting was performed to assess PPP3CB-GFP expression (left panel). Tubulin was used as a loading control. **(C)** Quantification of apoptotic rates was determined as in (C) (right panel). Data are presented as the mean ± SD, n = 4 biological replicates, **P < 0.01, *P < 0.05, unpaired t test. **(F)** Representative images of a clonogenic assay using PC9-GFP or PC9-PPP3CB-GFP cells cultured with osimertinib (osi, 0.03 μM) for 13 d.
Source data are available for this figure.

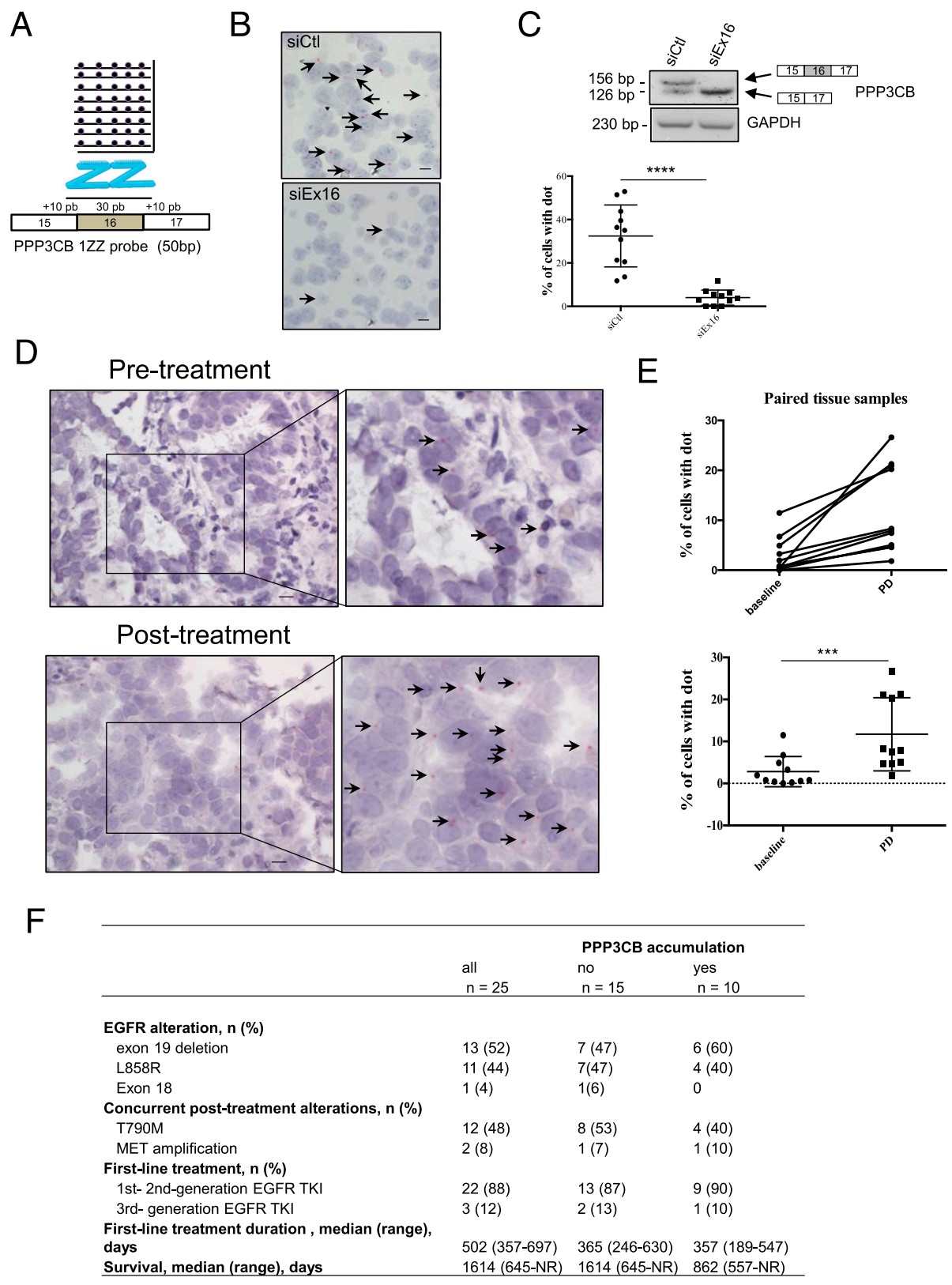

**Figure 2. Expression of *PPP3CB* mRNA in human non–small-cell lung cancer biopsies.**
Detection of *PPP3CB* transcript (pink dots) in PC9/OR cells using RNA BaseScope. **(A)** Minimal 50-bp 1ZZ probe containing complementary sequences to the 30 bp of exon 16 of *PPP3CB* plus 10 bp in exons 15 and 17 was designed according to the manufacturer's recommendations (Bio-Techne). **(B, C)** PC9/OR cells were transfected with control siRNA (siCtl) or with siRNA against exon 16 of *PPP3CB* for 72 h. **(B)** Cells were processed for RNA BaseScope with the *PPP3CB* 1ZZ probe. Arrows illustrate pink dots.

clones inhibits the anticancer effect of EGFR TKIs through activation of the MEK/ERK pathway.

Calcineurin is activated by increased Ca$^{2+}$ influx (Klee et al, 1998). As EGFR TKI treatment was required for CsA-induced apoptosis in resistant cells (Fig 3B), we wondered whether EGFR TKIs activate the calcineurin pathway by enhancing cytoplasmic Ca$^{2+}$ influx. To test this hypothesis, PC9/GR and PC9/OR cells were incubated with increasing concentrations of EGFR TKIs and labeled at different time points with the Ca$^{2+}$-sensitive fluorescence indicator, cell-permeant Fluo-4/AM, to monitor intracellular Ca$^{2+}$ levels. The results showed that intracellular Ca$^{2+}$ levels (iCa$^{2+}$) increase in a dose-dependent manner upon EGFR TKI treatment (Figs 4J and S3E). We also performed a direct assessment of Ca$^{2+}$ signals by microscopy imaging after Fluo-4/AM labeling. In both PC9/GR and PC9/OR cells, EGFR TKIs increased iCa$^{2+}$ signals (Figs 4K and S3F). Interestingly, Dougherty et al previously reported that when iCa$^{2+}$ levels increase, calcineurin contributes to the scaffolding function of KSR2 to promote ERK cascade activation and signaling (Dougherty et al, 2009). To investigate the role of KSR2 in our model, we used APS-2-79, a molecule that stabilizes KSR2 in an inactive state that antagonizes ERK phosphorylation (Dhawan et al, 2016). Co-treating resistant cells with EGFR TKIs and APS-2-79 decreased pERK levels compared with each single drug (Figs 4L and S3G). Furthermore, depleting KSR2 using siRNA increased apoptotic levels in the presence of EGFR TKIs, whereas no obvious effect was observed in the absence of EGFR TKIs (Fig S3H and I). These results are reminiscent of those obtained when calcineurin is inactivated, and strongly suggested that KSR2 mediates Ca$^{2+}$/calcineurin-dependent ERK activation.

### A CsA/trametinib/osimertinib combination of treatment inhibits the survival of the 2D and 3D PC9/OR cell culture

Although the combination of EGFR and MEK/ERK inhibitors has shown some clinical improvements in resistant EGFR-mutant NSCLC patients (Dagogo-Jack et al, 2019; Xie et al, 2021), the use of a tri-therapeutic osimertinib/trametinib/dabrafenib (BRAF inhibitor) treatment has recently demonstrated promising results in EGFR-mutant/BRAF V600E lung adenocarcinoma patients (Huang et al, 2019; Meng et al, 2020; Zhou et al, 2020; Ribeiro et al, 2021; Chimbangu et al, 2024). As our results provided a strong rationale for targeting the EGFR/calcineurin/MEK/ERK signaling pathway to overcome EGFR TKI resistance, we postulated that a tri-therapy combining subtoxic concentrations of each drug could provide a stronger antitumor benefit than bi-therapies. CsA and trametinib concentrations were thus defined so that each drug had a minimal effect on cell growth when used in bi-therapy with osimertinib. In 2D cell culture and MTS assays, the osimertinib/CsA/trametinib combo was highly effective in decreasing the survival of resistant PC9/OR cells (Fig 5A). A strong antitumor effect of the tri-therapy was also observed using colony formation assays as evidenced by a high cell growth inhibition compared with all other conditions (Fig 5B). To further assess treatment effectiveness, 3D cell culture experiments were performed. Again, the osimertinib/CsA/trametinib combo was more effective in decreasing the growth of the spheroids relative to control 3D structures (treatment with DMSO) or to single or dual treatments in the range of the concentrations used (Fig 5C and D). A strong toxic effect of the tri-therapy was observed as the viability of PC9/OR spheroids was greatly reduced as compared to control conditions (Fig 5E).

### Additive antitumor effect of combining osimertinib with trametinib and cyclosporin A in xenografts derived from an osimertinib-resistant PCR/OR cell line

Then, we tested the antitumor effect of the tri-therapy in xenografts derived from resistant PCR/OR cells. Looking at the mean tumor volume and the percentage of change in the tumor volume from baseline after 29 d of treatment, trametinib, cyclosporin A, and osimertinib as monotherapy had no significant antitumor effect in xenografts (Fig 6A–D). The combinations of cyclosporin A plus osimertinib or trametinib plus osimertinib induced a significant antitumor response compared with the control group (P = 0.0142 and P = 0.0106, respectively) (Fig 6C). Finally, the tri-therapy induced a strong antitumor response (P = 0.0002 versus the control group and versus cyclosporin A + osimertinib and cyclosporin A + trametinib), with 4 of 9 mice regressing (% of change < −0.5) and 4 of 9 stabilizing (% of change between −0.5 and +0.5) their tumors after 29 d of treatment (Fig 6D). This strong antitumor effect of the tri-therapy was also reflected in the probability of progression (two- to fourfold increase in RTV) (Fig S4A and B) and was dose-dependent (Fig S5). The triple combination was well tolerated by the mice as they showed no weight loss after 29 d of treatment (Fig S4C).

## Discussion

Resistance to EGFR TKI treatment is a major brake for long-term control of NSCLC with EGFR mutation in the clinic. Although the landscape of acquired resistance to EGFR TKIs, including the third-generation TKI such as osimertinib, reveals "on-target" mechanisms, "off-target" mechanisms, or phenotypic plasticity (Passaro et al, 2021; Blaquier et al, 2023), over half of the mechanisms remains unknown and requires additional exploratory analyses. An increased understanding of resistance mechanisms at the time of

Scale bars: 10 μm. **(C)** Quantification of RNA BaseScope data. Hematoxylin-counterstained tumor cells (blue) with or without pink dots (PPP3CB) were counted in 11 distinct fields within the slide (160 cells at least per condition). Results were expressed as % of cells with at least one dot. Data are presented as the mean ± SD, ****P < 0.0001, unpaired t test. Neutralization of the PPP3CB exon 16 transcript was checked by RT–PCR. GAPDH was used as a loading control. **(D)** Representative images of the RNA BaseScope assay in matched biopsies showing the increased expression of PPP3CB Ex16 mRNA levels (pink dots, arrows) in post-treatment sample. Scale bars: 10 μm. **(E)** Accumulation of PPP3CB Ex16 in 11 paired tumor biopsies from the global cohort of 43 patients (upper panel) (7–10 images within each section, mean = 491 total cells per sample). The statistical significance of PPP3CB Ex16 increases in the 11 paired samples (lower panel). PD, progression disease. ***P = 0.001, paired t test (Wilcoxon's test). **(F)** Mechanisms of acquired resistance in 25 paired biopsy subcohort of patients treated with first-line EGFR TKIs.
Source data are available for this figure.

**Table 1.  Patient/tumor characteristics and PPP3CB status.**

| | Full cohort | Subcohort with 1 or 2 lines of EGFR TKI | Subcohort with 1 or 2 lines of EGFR TKI + chemotherapy |
|---|---|---|---|
| | **n = 43** | **n = 30** | **n = 13** |
| Female sex, n (%) | | | |
| Female | 30 (70) | 22 (73) | 8 (62) |
| Male | 13 (30) | 8 (27) | 5 (38) |
| Age, median (range), years | 68 (63–74) | 68 (63–73) | 70 (61–77) |
| Histology, n (%) | | | |
| Adenocarcinoma | 41 (95) | 29 (97) | 12 (92) |
| Non-adenocarcinoma | 2 (5) | 1 (3) | 1 (8) |
| EGFR alteration at baseline, n (%) | | | |
| Exon 19 deletion | 22 (51) | 16 (53) | 6 (46) |
| L858R | 19 (44) | 13 (43) | 6 (46) |
| Exon 18 | 1 (2) | 1 (3) | 0 |
| Exon 19 duplication | 1 (2) | 0 | 1 (8) |
| First-line treatment, n (%) | | | |
| First- and second-generation EGFR TKI | 35 (81) | 27 (90) | 8 (62) |
| Third-generation EGFR TKI | 3 (7) | 3 (10) | 0 |
| Chemotherapy | 5 (12) | 0 | 5 (38) |
| No. of lines of treatment, n (%) | | | |
| One | 25 (58) | 25 (83) | 0 |
| Two | 13 (30) | 5 (17) | 8 (62) |
| Three | 5 (12) | 0 | 5 (38) |
| Delay between paired biopsies, median (range), days | 479 (315–701) | 436 (300–701) | 556 (377–641) |
| PPP3CB accumulation, n (%) | | | |
| No | 32 (74) | 19 (63) | 13 (100) |
| Yes | 11 (26) | 11 (37) | 0 |
| Survival, median (range), days | 944 (560-NR) | 1614 (645-NR) | 736 (518–944) |

Qualitative variables were reported as n (%) and quantitative variables as median (IQR 25–75%). IQR, interquartile range; NR, not reached.

progression after EGFR TKI treatment is essential to design new therapeutic strategies. Our results identify PPP3CB overexpression as a new candidate mechanism of acquired resistance to EGFR TKI, and demonstrate that targeting the PPP3CB-dependent signaling pathway is an effective strategy to overcome resistance to osimertinib.

Our results showed that a *PPP3CB* splice variant, which retains Ex16 and encodes the full-length catalytic subunit of calcineurin, accumulates in lung tumor cells with acquired resistance to first-, second-, or third-generation EGFR TKIs. These results were validated in a retrospective cohort of *EGFR*-mutant NSCLC patients who received a single line of EGFR TKI including osimertinib, with 20% of post-treatment samples having *PPP3CB* mRNA accumulation without any known genetic mechanism of acquired resistance. The baseline expression of *PPP3CB* mRNA was heterogeneous in pre-treatment biopsies, and a tendency for patients with a low level to have a better response to EGFR TKI was observed. Analysis of larger clinical cohorts should evaluate the importance of this pre-

treatment modification for the clinical outcome of NSCLC patients who receive EGFR TKI in the front line. This is of particular interest as it could guide for decision-making regarding the treatment strategy in the first-line setting.

In resistant *EGFR*-mutant NSCLC cell lines, we demonstrated that PPP3CB up-regulation reduces sensitivity to EGFR TKIs by activating the MEK/ERK signaling pathway. Alterations in *MET*, *HER2*, *RAS*, and *RAF* genes have all been reported to induce EGFR-independent activation of Ras/Raf/MEK/ERK signaling in EGFR TKI– resistant cells (Blaquier et al, 2023). Therefore, our results identify PPP3CB overexpression as a new mechanism of Ras/Raf/MEK/ERK pathway reactivation associated with EGFR TKI resistance. We showed that treatment with EGFR TKIs is required for CsA-induced ERK inactivation. This suggested that EGFR TKIs regulate calcineurin signaling. As calcineurin responds to increased concentrations of calcium in the cell (Klee et al, 1998; Rusnak & Mertz, 2000), we quantified calcium influx in resistant

**Table 2. Clinical impact of basal PPP3CB expression on first-line EGFR TKI.**

| | Full cohort | Basal PPP3CB < 1.9 | Basal PPP3CB > 1.9 |
|---|---|---|---|
| | n = 25 | n = 15 | n = 10 |
| First-line treatment, n (%) | | | |
| First- and second-generation EGFR TKI | 22 (88) | 13 (87) | 9 (90) |
| Third-generation EGFR TKI | 3 (12) | 2 (13) | 1 (101) |
| Response[a] | | | |
| Partial response | 17 (68) | 11 (73) | 6 (60) |
| Stable disease | 8 (32) | 4 (27) | 4 (40) |
| First-line treatment duration, median (range), days[b] | 502 (357–697) | 599 (365–856) | 277 (300–547) |

Median baseline PPP3CB level = 1.9 (IQR 25–75%, 0.8–4.9).
[a]No patient progressed at first tumor evaluation in this cohort.
[b]First-line treatment was stopped for all patients.

cells and found that EGFR TKIs increase intracellular levels of $Ca^{2+}$ in a dose-dependent manner. Therefore, our data support a model whereby EGFR TKIs increase $iCa^{2+}$ levels and activate a calcineurin-dependent MEK/ERK pathway. We observed that EGFR TKIs also increase intracellular calcium levels in sensitive PC9 cells (data not shown) although phospho-ERK is inhibited. These results further support the role of PPP3CB accumulation in driving resistance through reactivation of the MEK/ERK pathway in response to EGFR TKIs. Using various assays, we could show that inactivation/neutralization of KSR2 diminishes phospho-ERK levels and restores apoptosis induction in resistant cells treated with EGFR TKI. These results strongly suggested that KSR2 mediates $Ca^{2+}$/calcineurin-dependent ERK activation. Liu et al previously reported that a $Ca^{2+}$/calcineurin/KSR2/ERK pathway contributes to primary crizotinib resistance in *MET*-amplified NSCLC cells (Liu et al, 2019). In this study, we identify dysregulation of calcineurin expression (by the mean of PPP3CB overexpression) as a critical node for activation of the KSR2/ERK pathway by EGFR TKI, and demonstrate that activation of this pathway is a mechanism of acquired resistance of EGFR-mutant NSCLC cells to EGFR TKI.

Tackling resistance to EGFR TKIs is a critical challenge in a clinical setting. Bi-therapies involving osimertinib and MEK TKI have shown antitumor activity in clinical case studies of EGFR-mutant NSCLC patients (Dagogo-Jack et al, 2019; Xie et al, 2021), and some clinical trials are in progress to evaluate the efficacy of the osimertinib/selumetinib (MEK inhibitor) combo (NCT03944772, NCT03392246). Although we showed that osimertinib/trametinib and osimertinib/cyclosporin A combinations inhibit the growth of resistant cells with PPP3CB overexpression, our results provide a strong rationale for the use of a osimertinib/trametinib/cyclosporinA tri-therapy. Indeed, the triple combination achieved a much stronger antitumor response than the bi-therapies, with 44% of regressing tumors. No apparent toxicity of the triple combination was observed with regard to weight loss. Concomitant administration of EGFR TKI erlotinib and CsA has been used in a liver transplant recipient with EGFR-mutant NSCLC without added toxicity (De Pas et al, 2009). Although MEK TKIs and EGFR TKIs have overlapping toxicity (Satoh et al, 2020), recent results from the TATTON trial reported that an appropriate dosing schedule allows tolerable toxicity in EGFR-mutated NSCLC (Yang et al, 2022). Importantly, in EGFR-mutant/BRAF V600E NSCLC patients, the dabrafenib/trametinib/osimertinib triple combination therapy has demonstrated manageable toxic side effects (Huang et al, 2019; Meng et al, 2020; Zhou et al, 2020; Ribeiro et al, 2021; Chimbangu et al, 2024). Therefore, we postulate that combining EGFR, calcineurin, and MEK inhibitors is a promising therapeutic strategy for treatment of EGFR-mutant NSCLC patients that accumulate PPP3CB after progression on initial EGFR TKI.

# Materials and Methods

### Cell lines and reagents

PC9 cells were provided by Dr. A. Gazdar (UT Southwestern) and recently authenticated by DNA STR profiling (ATCC Cell Line Authentication Service, LGC Standards). Gefitinib- and dacomitinib-resistant cell lines were newly established in our laboratory by exposing PC9 cells to gradually increasing concentrations of TKI as previously reported (Hatat et al, 2022). Single-cell clones were derived by limiting dilution, and further study was performed on resistant clones that did not carry common mechanisms of resistance. We established PC9 cells with acquired resistance to osimertinib by incubating the parental cells for 4 mo with increased concentrations of osimertinib and isolated surviving cells (Fig S6). The osimertinib-resistant clones PC9/OR1 to OR3 did not carry known genetic mechanisms of resistance (*MET* and *HER2* amplification, EGFR secondary mutations, KRAS, BRAF, HER2, or PIK3CA mutation) as detected by next-generation sequencing analysis using the panel "OST DNA kit combo EA" on an Ion GeneStudio S5 Plus sequencer (Thermo Fisher Scientific) as recommended by the manufacturer. All clones maintained resistance to EGFR TKI even after withdrawal of the drugs from the culture medium. The absence of mycoplasma was routinely tested (MycoAlert Mycoplasma Detection Kit). Gefitinib, dacomitinib, osimertinib, trametinib, cyclosporin A, and APS-2-79 were purchased from TargetMol (Euromedex).

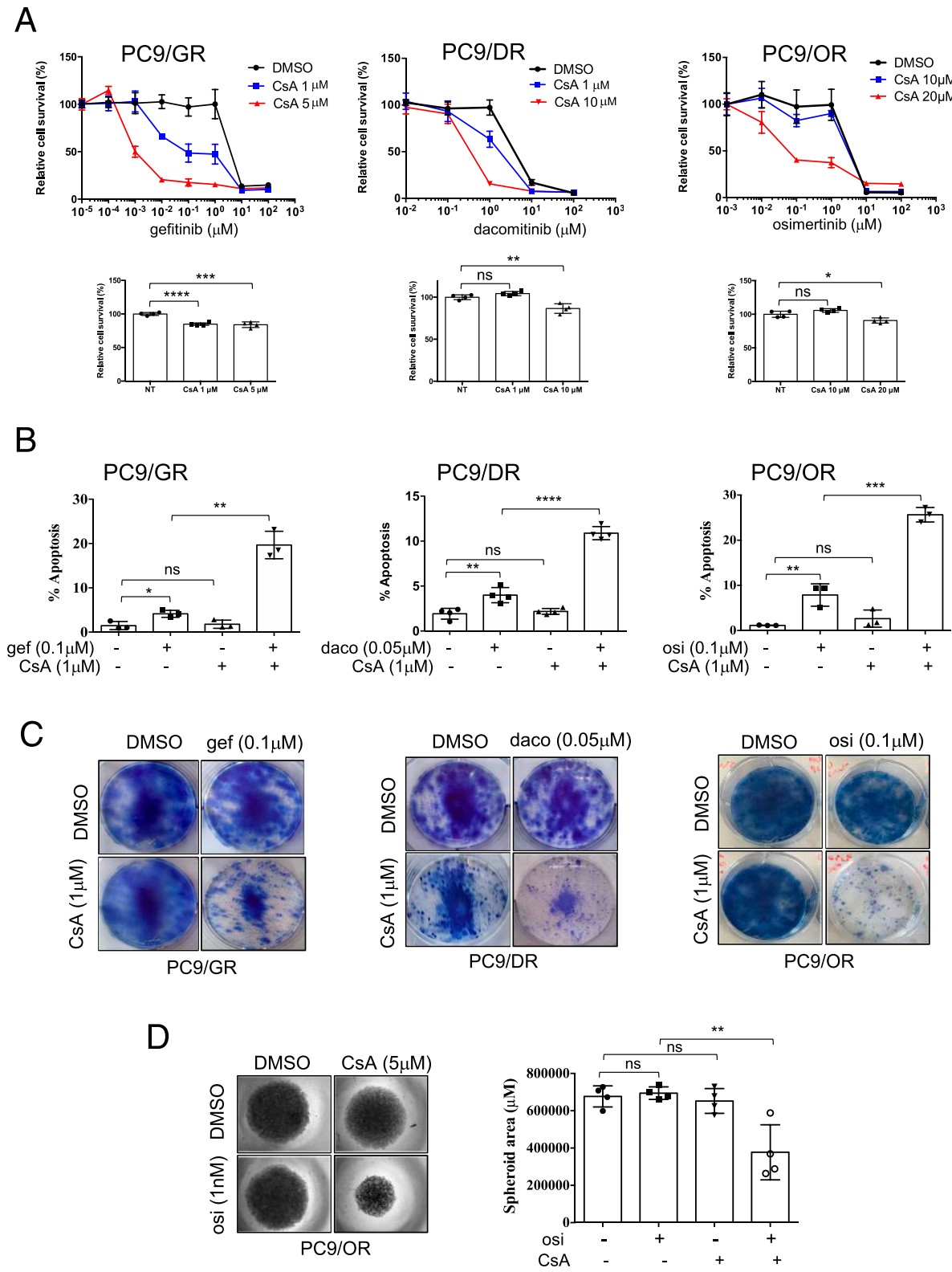

**Figure 3. CsA sensitizes resistant cells to EGFR TKI treatment.**
**(A, B, C, D)** Cells were cultured with indicated EGFR TKIs in the presence or absence of CsA for 96 h (A), 72 h (B), 2 wk (C), or 10 d (D). Concentrations used for each drug in each assay are indicated. **(A)** Cell viability was measured by an MTS assay (upper panel). CsA toxicity is shown in lower panels. Data are presented as the mean ± SD, n = 4 biological replicates. ****$P < 0.0001$, ***$P < 0.001$, **$P < 0.01$, *$P < 0.05$, ns, not significant. **(B)** Apoptosis was quantified by active caspase 3 staining and flow cytometry. Data are presented as the mean ± SD, n = 3 or 4 biological replicates. ****$P < 0.0001$, ***$P < 0.001$, **$P < 0.01$, *$P < 0.05$, ns, not significant, unpaired $t$ test. **(C)** Representative

## Two-dimensional (2D) and three-dimensional (3D) Cell culture

Cells were grown in RPMI 1640 + GlutaMAX medium (Gibco) supplemented with 10% FBS and kept at 37°C in a humidified atmosphere containing 5% $CO_2$. Spheroids were generated by plating 1,000 cells/well into 96-well round-bottom ultra-low attachment (ULA) spheroid microplates (Corning). The spheroid culture was performed in complete medium in a humidified atmosphere with 5% $CO_2$. Spheroid formation and growth were assessed by microscopic examination using an inverted microscope and by imaging the spheroids at each time point.

## Tumor samples and clinical data

This study was conducted in accordance with the provisions of the Declaration of Helsinki and good clinical practice guidelines after approval by the Ethics Committee of Clinical Investigation Centers of Rhone Alpes Auvergne (IRB 5891; Clermont-Ferrand). All living patients freely provided informed written consent. Tumor samples were collected at Grenoble-Alpes University Hospital (Grenoble, France), Daniel Hollard Institute (Grenoble, France), Annecy Genevois Hospital (Annecy, France), Alpes Leman Hospital (Contamine-sur-Arve, France), and the Institut du Thorax Curie-Montsouris (Paris, France). All NSCLC samples underwent standard pathological examination and had a molecular biology analysis with next-generation sequencing at diagnosis and during the follow-up.

## Cell viability assays

Cell viability assays were conducted on 96-well plates (5,000 cells/well for 2D culture and 1,000 cells/well for spheroids). 2D cultured cells were grown for 24 h before treatment with drugs for 96 h. Cell viability was assessed using CellTiter 96 AQueous One Solution Cell Proliferation Assay (Promega). EC50 values were calculated using GraphPad Prism version 6. In siRNA experiments, cells were transfected with siRNA and treated with EGFR TKI for 72 h before cell viability was analyzed. For 3D culture, cells were treated 4 d (d4) after plating, once spheroid structures could be easily visualized. Treatment was repeated once at day 11 and was stopped at day 14 (d14). Spheroid growth was evaluated at days 4, 7, 11, and 14 by calculating spheroid area (ImageJ software) after microscopic acquisition (MetaMorph 7812). Cellular viability was assessed at days 4 and 14 using CellTiter-Glo 3D Cell Viability Assay (Promega) and a luminescent plate reader (CLARIOstar Plus; Bio-Rad).

## Colony formation assays

Cells (500 cells per well) were seeded onto six-well plates, allowed to adhere overnight, and treated with drugs for 7 d. Next, the culture medium was replaced with fresh RPMI containing 10% FBS and drugs, and cultured for an additional 7 d. For studies using PC9 cells overexpressing PPP3CB, cells (100,000 cells per well) were seeded onto 24-well plates and the culture medium was replaced with fresh culture medium containing drugs every 3 d until 6 d of drug treatment. Then, the medium was removed and cells were washed twice with PBS, fixed with formol 4% for 5 min, washed once with PBS, stained with 1% methylene blue in borate buffer for 15 min, washed with tap water, and imaged.

## Western blot analysis

Cell lysis was performed in RIPA buffer as described previously (Hatat et al, 2022). The expression of the proteins of interest was assessed by Western blotting using *GAPDH* or *tubulin* as housekeeping genes for normalization. GFP (B-2, sc-9996), PARP (sc-8007), GAPDH (sc-47724), and tubulin (B-5-1-2, sc-23948) antibodies were purchased from Santa-Cruz Technology. p44/42 MAPK Erk1/2 (137F5, 4695), p44/42 MAPK (T202/Y204) (D13.14 4E, 4370), cleaved caspase 3 D175 (9661), AKT (9272), pAkt (Ser473) (D9E, 4060), Stat3 (79D7, 4904), pStat3 (Tyr705) (D3A7, 9145), EGFR (D38B1, 4267), and pEGFR (Y1068) (D7A5, 3777) were obtained from Cell Signaling Technologies.

## RNA BaseScope

RNA BaseScope was performed using the BaseScope Reagent Kit v2 and BA-Hs-PPP3CB-1zz-st probe (Bio-Techne). The probe design included the 30 bp of exon 16 of *PPP3CB* and the adjacent 10 bp on either side (10 bp from exon 15 and 10 bp from exon 17) to obtain the minimal 50-bp hybridization sequence required for further processing. The 3-$\mu$M FFPE tumor sections were processed according to the manufacturer's protocol with modified retrieval incubation time (25 min) and AMP7 amplification step (45 min). Sections were imaged (7–10 images within each section, mean 491 total cells analyzed per sample) using Axio Imager M2 (Carl Zeiss) with a 40x lens and AxioVision software. PPP3CB was detected as pink dots within the cells, which were counterstained in blue (hematoxylin followed by ammonia). Analysis of the sections was carried out using ImageJ software, and the total number of blue cells and pink dots was counted for each sample. Results were expressed as the percentage of cells with dots.

## Measurement of intracellular calcium concentration

The intracellular calcium concentration was measured using a $Ca^{2+}$-sensitive fluorescence indicator, cell-permeant Fluo-4/AM (Thermo Fisher Scientific). Cells were seeded onto 96-well plates (10,000 cells per well) or 24-well plates (30,000 cells per well) and treated 24 h later with drugs for various times. Then, cells were washed with Hank's balanced salt solution (HBSS), incubated for 60 min at 37°C in dark in HBSS containing 3 µM Fluo-4/AM and 0.1% of Pluronic F-127, washed with HBSS, and incubated

---

images of a clonogenic assay. **(D)** Representative images of PC9/OR spheroids cultured with osimertinib (osi) and/or CsA. Data are presented as the mean ± SD, n = 4 biological replicates; two technical replicates for each biological replicate. **P < 0.01, ns, not significant, unpaired *t* test.
Source data are available for this figure.

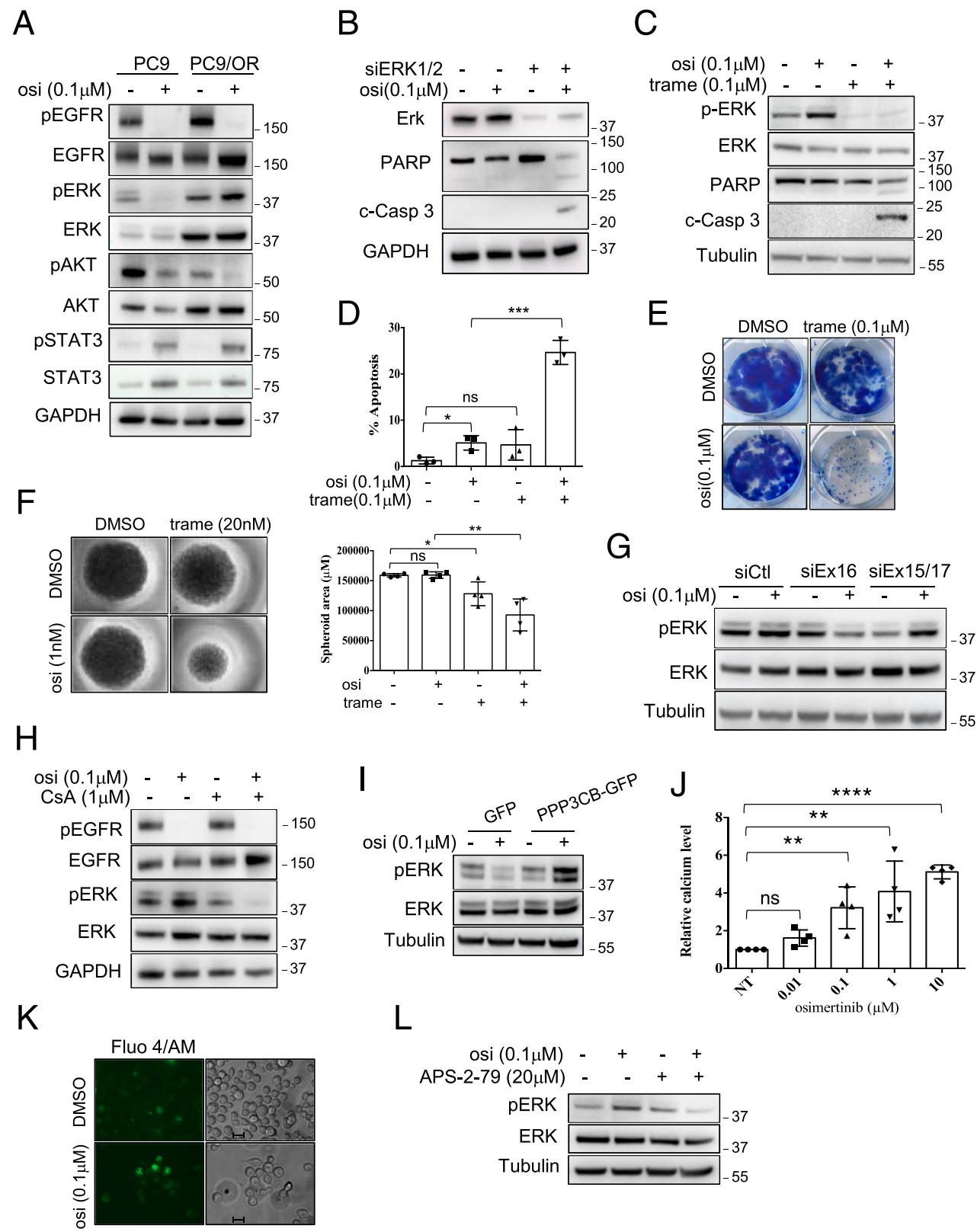

**Figure 4. Osimertinib activates a Ca²⁺/calcineurin/MEK/ERK1/2 pathway that prevents apoptosis in response to osimertinib.**
**(A)** Western blot analysis of proteins of the EGFR pathway in cells treated with osimertinib (osi) for 72 h. **(B)** Representative immunoblots of indicated proteins in PC9/OR cells transfected with siRNA against *ERK1/2* (siERK1/2) or with control siRNA (−) and cultured (+) or not (−) with osimertinib (osi) for 72 h. c-Casp3 = cleaved caspase 3. **(C, D)** PC9/OR cells were cultured with osimertinib (osi) and/or trametinib (trame) for 72 h. **(C)** Representative immunoblots of indicated proteins. **(D)** Quantification of apoptotic

in dark for an additional 30 min at 37°C. Data were acquired using a microplate reader (CLARIOstar Plus; Bio-Rad) with the excitation and emission filter at 488 and 520 nm, respectively, or cells were imaged with Axio Observer Z1 (Carl Zeiss). For microplate experiments, protein concentration was further evaluated in each condition using Pierce BCA Protein Assay Kit (Thermo Fisher Scientific). A fluorescence intensity/protein concentration ratio was calculated for each well. Relative Fluo/protein concentration ratios were represented as a fold increase relative to untreated cells, which were arbitrarily assigned to 1.

### cDNA constructs and lentiviral transduction

The *PPP3CB* plasmid is the result of C-terminal fusion of GFP to PPP3CB. This plasmid was constructed from a HINDIII-digested pSico R CAG/GFP backbone and *PPP3CB* sequence (Transcript variant 2, NM_021132.4) amplified from a pcDNA3.1+/C-(K)-DYK-PPP3CB vector (GenScript) by PHUSION High-Fidelity DNA Polymerase (NEB) using the Gibson assembly following the supplier's instructions. Lentivirus production and cell infection were performed using pSico-PPP3CB-GFP and pSico-GFP as a control as described previously (Hatat et al, 2022). Cells were FACS-sorted (Aria IIu; BD Biosciences) based on the GFP expression level, and stable cell lines were amplified for further use.

### Transfection of siRNA

Transfection was performed using jetPRIME (Ozyme) or RNAiMAX (Invitrogen; Thermo Fisher Scientific) according to the manufacturer's protocol, and cells were analyzed 72 h post-transfection. The siRNAs used were the following: PPP3CB Ex16 5′-AAGUGCCACAGUU-GAGGCUAUUGA-3′; PPP3CB Ex15/17 5′-CCUGCAAAGUGCAAUACGAGG-3′; ERK 5′-CCAGGAUACAGAUCUUAAA-3′; KSR2 5′-GAA-CAA-GAA-GAG-AGA-CCU-ACC-3′. For all RNA interference experiments, the siRNA used as a control was 5′-UCGGCUCUUACGCAUUCAA-3′.

### RNA extraction, RT–PCR, and qPCR

Total RNA was extracted using the NucleoSpin RNA isolation kit (Macherey-Nagel) and reverse-transcribed with iScript RT Supermix (Bio-Rad). PCR was carried out using GoTaq (Promega) and the following primers: PPP3CB Ex16-Fw 5′-CTGACTCCCACAGGGATGTT-3′, PPP3CB Ex16-Rv 5′-ATCCAAACCCTTTGCCTCTT-3′; ACOT9 Ex5-Fw 5′-CAGCTTACTCCTGGAAGA-3′, ACOT9 Ex7-Rv 5′-GTGGATGCTCCTACTATC-3′;

CD46 Ex12-Fw 5′-CGTACAGATATCTTCAAAGG-3′, CD46 Ex14-Rv 5′-TGATTTAGTCTGGTAAGT-3′; ESYT2 Ex15-Fw 5′-CTGACAAAGACCAAGCCAAC-3′, ESYT2 Ex17-Rv 5′-CCAACTGACATCTGGACAAC-3′; RAB17 Ex3-Fw 5′-AAGTCCAGCTTGGCTCTTCG-3′, RAB17 Ex5-Rv 5′-GCTGAGGTCCGTCTTGTTGC-3′; ARFGAP2 Ex7-Fw 5′-CCAACACAGACCTGCTTGG-3′, ARFGAP2 Ex9-Rv 5′-CTTAGCTGCTGCTGGCTTC-3′; GK Ex21-Fw 5′-CTGGGTTACAACTCAATCTCC-3′, GK Ex23-Rv 5′-GGAATCCATGAGTTGGTAGG-3′; SMC5 Ex18-Fw 5′-GGTGCA-GAGCAGACTCTTCC-3′, SMC5 Ex20-Rv 5′-AGGATTCAGTCCCGTGAAGC-3′; GAPDH-Fw 5′-CGA-GAT-CCC-TCC-AAA-ATC-AA-3′, GAPDH-Rv 5′-ATC-CAC-AGT-CTT-CTG-GGT-GG-3′. PCR products were analyzed on 2% agarose gels. Quantitative PCR was performed using iTaq Universal SYBR (Bio-Rad) and the following primers: KSR2-Fw 5′-GGCAACCTTTCCAAACAAGAC-3′, KSR2-Rv 5′-GGCTGGTGTGACAATAGTGC-3′.

### Apoptosis detection

Apoptosis was evaluated by the PE-conjugated monoclonal active caspase 3 antibody apoptosis kit (Pharmingen, BD Biosciences) or FITC Active Caspase-3 Apoptosis Kit (Pharmingen, BD Biosciences) according to the manufacturer's instructions and analyzed by fluorescence-activated cell sorting (FACS, Accuri C6, BD Biosciences or Attune, Life Technologies) using Accuri C6 software (BD Biosciences) or FCS Express 7 software (DeNovo Software).

### In vivo studies

8-wk-old female Swiss *Nude* mice were purchased from Charles River Laboratories and maintained under specific pathogen-free conditions. The care and animal housing were in accordance with institutional guidelines and with the recommendations of the French Ethics Committee (Comité Ethique en matière d'expérimentation animale de l'Institut Curie, national registration number: #118, authorization no.: APAFIS#25909-2020060410487032 v2). The housing facility was kept at 22°C (±2°C) with a relative humidity of 30–70%. The light/dark cycle was 12-h light/12-h dark. First, PCR/OR3 cells ($6.10^6$ cells/0.1 ml) were injected subcutaneously into a Swiss *Nude* mouse. When the tumor reached 1,500 mm³, it was excised, cut into small fragments, and re-implanted into 80 Swiss *Nude* mice. Animals were randomly divided (n = 7–10 per group) when tumor volume reached 100–200 mm³. Cyclosporin A (Neoral, Novartis), osimertinib (Tagrisso, AstraZeneca), and trametinib (MedChemExpress) were administrated orally 4 d per week (D1-D2-D4-D5) at a daily dosage of 25 mg/kg, 5 mg/kg, and

rates was determined by active caspase 3 staining and flow cytometry. Data are presented as the mean ± SD, n = 3 biological replicates. ***$P < 0.001$, *$P < 0.05$, ns, not significant, unpaired *t* test. **(E)** Representative images of the colony formation assay in PC9/OR cells cultured with osimertinib (osi) and/or trametinib (trame) for 2 wk. **(F)** Representative images of PC9/OR spheroids cultured for 10 d with osimertinib (osi) and/or trametinib (trame). Data are presented as the mean ± SD, n = 4 biological replicates; two technical replicates for each biological replicate. **$P < 0.01$, *$P < 0.05$, ns, not significant, unpaired *t* test. **(G)** Representative immunoblots of pERK/ERK in PC9/OR cells transfected with *PPP3CB* siEx16 or siEx15/17 or control siRNA (siCtl) and treated with osimertinib (osi) for 72 h. **(H)** Representative immunoblots of pERK/ERK in PC9/OR cells cultured with osimertinib (osi) and/or CsA for 72 h. **(I)** Representative immunoblots of pERK/ERK in PC9/GFP and PC9/PPP3CB-GFP cells cultured with osimertinib (osi) for 18 h. **(J, K)** Measurement of iCa2+ using the fluorescence indicator, cell-permeant Fluo-4/AM. **(J)** Dose-dependent effect of osimertinib on the $iCa^{2+}$ level in PC9/OR cells treated for 48 h. ns, not significant. Data are presented as the mean ± SD, n = 4 biological replicates; two technical replicates for each biological replicate. ****$P < 0.0001$, **$P < 0.01$, unpaired *t* test. **(K)** Live-cell imaging of $iCa^{2+}$ using epifluorescence microscopy (green signal) in PC9/OR cells cultured for 48 h with osimertinib (osi). Bar = 10 $\mu m$. **(L)** Representative immunoblots of pERK/ERK in PC9/OR cells cultured with osimertinib (osi) and/or APS-2-79 for 72 h. Data, GAPDH, or tubulin was used as a loading control. The concentration used for each drug in each assay is indicated. **(A, B, C, G, H, I, L)** Three biological replicates. Source data are available for this figure.

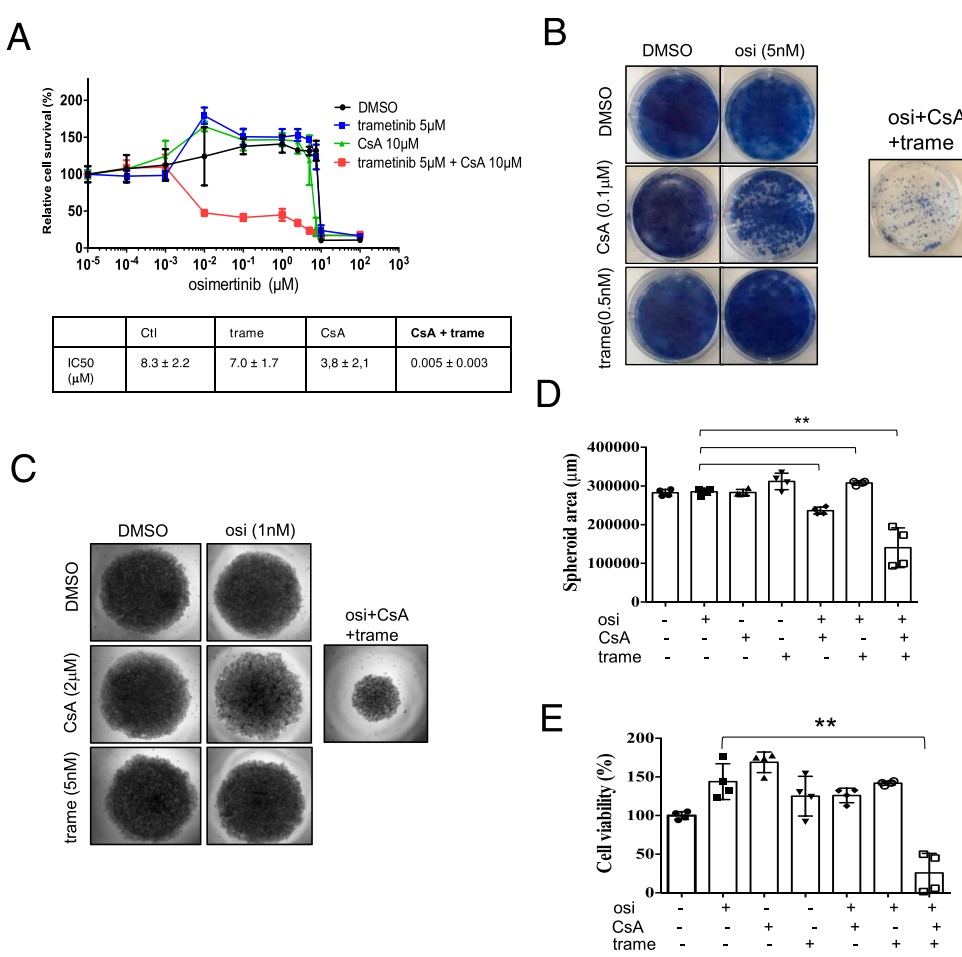

**Figure 5. Combined CsA/trametinib treatment restores sensitivity to osimertinib in 2D and 3D cell culture assays. (A)** Representative MTS cell viability assays in PC9/OR cells treated with osimertinib, CsA, and trametinib. Data are presented as the mean ± SD, n = 4 biological replicates; three technical replicates for each biological replicate. IC50 values were calculated using GraphPad Prism. Data are presented as the mean ± SD. **(B)** Representative images of a clonogenic assay in PC9/OR cells cultured with osimertinib (Osi) and/or CsA and/or trametinib (trame) for 2 wk. **(C, D, E)** 3D culture of PC9/OR cells. **(C)** Representative images of PC9/OR spheroids cultured for 10 d with osimertinib (osi), CsA, and/or trametinib (trame). **(D)** Spheroid area, mean ± SD, **P < 0.01, unpaired t test, and (E) cell viability, mean ± SD, **P < 0.01, unpaired t test, were assessed after 10 d of drug treatment. **(D, E)** n = 4 biological replicates; two technical replicates for each biological replicate. In all experiments, the concentration used for each drug is indicated. Source data are available for this figure.

0.1 mg/kg, respectively. Tumor growth was evaluated by measuring two perpendicular tumor diameters with a manual caliper twice per week. Individual tumor volumes were calculated as $V = a × b^2/2$, where a is the largest diameter, and b is the smallest diameter. For each tumor, the percent change in volume was calculated as $[(V_f − V_0)/V_0] × 100$, $V_0$ being the initial volume (at the beginning of treatment) and $V_f$ the final volume (at the end of treatment). The quality of the antitumor response was determined as follows: a decrease in tumor volume of at least 50% was classified as a huge or complete response, a volume change between −50% and +50% was considered as stable disease, and an increase in tumor volume of at least 50% was identified as progressive disease. The statistical analysis of tumor growth inhibition was performed with the Mann–Whitney test.

### Statistics

Analyses were performed using GraphPad Prism version 6 (GraphPad Software Inc.). In vitro data and xenograft tumor progression are presented as the mean ± SD. Statistical comparisons between two groups were conducted with a paired or an unpaired t test. A P-value less than 0.05 was considered statistically significant.

## Supplementary Information

## Acknowledgements

We thank Dr Nathalie Sturm and Bastien Marlot for the management of tumor samples. Imaging/flow cytometry experiments were done on Microcell Core Facility of the Institute for Advanced Biosciences (UGA—Inserm U1209—CNRS 5309). This facility belongs to the IBISA-ISdV platform, a member of the national infrastructure France-BioImaging supported by the French National Research Agency (ANR-10-INBS-04). We thank Solene Dufour for her assistance with Attune technology. We also thank the animal facility platform of the Institut Curie, as well as Wesnie Etienne, Laure Lagaye, and Harry Ahnine who worked in the Laboratory of Preclinical Investigation. This work received support from the "Institut National de la Santé et de la Recherche Médicale" U1209, the "Fondation ARC pour la recherche sur le cancer" (PJA 20181207847), the Ligue Nationale Contre le Cancer (comité de l'Allier et de l'Isère), the Association Espoir, GEFLUC Grenoble Dauphiné Savoie, PFIZER, AGIR pour les maladies chroniques, the ERiCAN program of Foundation MSD-Avenir supported by Canceropole CLARA (DS-2018-0015), and the "Institut National du cancer" (INCa PLBIO2020-115). N Zubchuk was supported by the ERiCAN program of Foundation MSD-Avenir and the "Institut National du cancer" (INCa PLBIO2020-115).

Figure 6. **In vivo efficacy of the cyclosporin A/trametinib/osimertinib combination in reversing acquired resistance to osimertinib.**
**(A)** Mean tumor volume (± SD) of the PC9/OR3 cell line model treated with cyclosporin A (25 mg/kg, 4x/w, PO) ± trametinib (0.1 mg/kg, 4x/w, PO) ± osimertinib (5 mg/kg, 5x/w, PO). n = 8–10 animals per group. **(B)** Individual tumor growth curves. n = 8–10 animals per group. **(C)** Statistical analysis of the tumor volume performed by an unpaired t test (Mann–Whitney test). n = 8–10 mice per group. **(D)** Percentage of volume change from initial volume after 29 d of treatment, percentages in gray correspond to a percentage of change lower than 50%, and percentages in black correspond to a percentage of change lower than −50%. n = 8–10 animals per group. Source data are available for this figure.

## Author Contributions

S Gazzeri: conceptualization, data curation, formal analysis, supervision, funding acquisition, investigation, methodology, project administration, and writing—original draft, review, and editing.
N Zubchuk: formal analysis and investigation.

E Montaudon: formal analysis, investigation, and methodology.
F Nemati: formal analysis, investigation, and methodology.
S Huot-Marchand: formal analysis and investigation.
G Berardi: resources and formal analysis.
A Pucciarelli: formal analysis and investigation.
Y Dib: formal analysis and investigation.

D Nerini: formal analysis and investigation.

C Oddou: resources.

M Pezet: formal analysis and investigation.

L David-Boudet: formal analysis and investigation.

C Ardin: resources.

F de Fraipont: resources, formal analysis, and investigation.

A Maraver: resources and writing—review and editing.

N Girard: resources, funding acquisition, and writing—review and editing.

D Decaudin: formal analysis, funding acquisition, methodology, and writing—review and editing.

A-C Toffart: resources, data curation, formal analysis, funding acquisition, project administration, and writing—review and editing.

B Eymin: conceptualization, funding acquisition, and writing—review and editing.

## Conflict of Interest Statement

N Girard has received research grants/support from AbbVie, Amgen, AstraZeneca, BeiGene, Boehringer Ingelheim, Bristol Myers Squibb, Daiichi-Sankyo, Gilead, Hoffmann-La Roche, Janssen, Leo Pharma, Lilly, Merck Serono, Merck Sharp & Dohme, Novartis, Sanofi, and Sivan; has provided consultative services for AbbVie, Amgen, AstraZeneca, BeiGene, Bristol Myers Squibb, Daiichi-Sankyo, Gilead, Ipsen, Hoffmann-La Roche, Janssen, Leo Pharma, Lilly, Merck Sharp & Dohme, Mirati, Novartis, Pfizer, Pierre Fabre, Sanofi, and Takeda; has participated on a data safety monitoring board for Hoffmann-La Roche; and has a family member employed with AstraZeneca. A-C Toffart has received consulting fees, payment, or honoraria for lectures and presentations from Astra Zeneca, BMS, Janssen, MSD, Pfizer, Sanofi, Roche, and Takeda; and support for attending meetings and/or travel from Astra Zeneca, Pfizer, MSD, and Roche. G Berardi received support for attending meetings and/or travel from Pfizer.

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
