## [Reviewer comments · Life Science Alliance]

Life Science Alliance

PPP3CB overexpression mediates EGFR-TKI resistance in lung tumors via calcineurin/MEK/ERK signaling

Sylvie Gazzeri, Nadiia Zubchuk, Elodie Montaudon, Fariba Némati, Sarah Huot-Marchand, Giulia Barardi, Amelie Pucciarelli, Yassir Dib, Dylan Nerini, Christiane Oddou, Mylène Pezet, Laurence David-Boudet, Camille Ardin, Florence de Fraipont, Antonio Maraver, Nicolas Girard, Didier Decaudin, Anne-Claire Toffart, and Béatrice Eymin

DOI: <https://doi.org/10.26508/lsa.202402873>

Corresponding author(s): Sylvie Gazzeri, INSERM U1209 and Béatrice Eymin, INSERM U1209

Review Timeline:

Submission Date:	2024-06-06
Editorial Decision:	2024-06-07
Revision Received:	2024-09-19
Editorial Decision:	2024-09-19
Revision Received:	2024-09-23
Accepted:	2024-09-24

Transaction Report:

June 7, 2024

Re: Life Science Alliance manuscript #LSA-2024-02873-T

Dr. Sylvie Gazzeri
INSERM U1209
AllÃ©{copyright, serif}e des Alpes
site SantÃ©{copyright, serif}
Grenoble 38000
FRANCE

Dear Dr. Gazzeri,

Thank you for submitting your manuscript entitled "Calcineurin confers resistance to EGFR TKI in Lung Cancer by activating MEK/ERK signalling pathway" to Life Science Alliance. We invite you to submit a revised manuscript addressing the following Reviewer comments:

- Address Reviewer 2's Major concerns #3-13 & 15.
- Address Reviewer 3's comments.

Thank you for this interesting contribution to Life Science Alliance. We are looking forward to receiving your revised manuscript.

Sincerely,

- A letter addressing the reviewers' comments point by point.
- An editable version of the final text (.DOC or .DOCX) is needed for copyediting (no PDFs).
- High-resolution figure, supplementary figure and video files uploaded as individual files: See our detailed guidelines for preparing your production-ready images, <https://www.life-science-alliance.org/authors>
- Summary blurb (enter in submission system): A short text summarizing in a single sentence the study (max. 200 characters including spaces). This text is used in conjunction with the titles of papers, hence should be informative and complementary to the title and running title. It should describe the context and significance of the findings for a general readership; it should be written in the present tense and refer to the work in the third person. Author names should not be mentioned.
- By submitting a revision, you attest that you are aware of our payment policies found here: <https://www.life-science-alliance.org/copyright-license-fee>

B. MANUSCRIPT ORGANIZATION AND FORMATTING:

Address Reviewer 2's Major concerns #3-13 & 15**Major concerns :****3. It is unclear what was the rationale to choose PPP3CB splicing and not the other ten altered splicing events that were identified by RNAseq**

Among the 11 exon skipping events that were validated by RT/PCR in PC9/GR and PC9/DR clones, 8 were also found in PC9/OR clones compared to parental PC9 cells. Of these 8 splicing events, only 2 (SMC5 and PPP3CB) were clearly observed **in all PC9/OR1-3, PC9/GR1-3 and PC9/DR1-3 clones** based on the calculated PSI (Fig 1A and Fig S1). We decided to focus on PPP3CB because several studies in the literature reported its pro-tumorigenic role in cancers and showed that calcineurin inhibitors could have potent anti-tumor effects in various cancer (Kawahara et al., 2015a, Kawahara et al., 2015b, Medyouf et al., 2007, Siamakpour-Reihani et al., 2011). This is documented on page 6 of the revised manuscript.

4 and 5. Since most of the conclusions were made based on mRNA levels, it would be valuable to show the protein levels of calcineurin, both in vitro and in patients samples. It is important to demonstrate that PPP3CB exon 16 splicing correlates with the increased calcineurin protein levels since the authors claim that only the exon 16 splice variant encodes the full-length isoform of PPP3CB

Western blot analysis of PPP3CB protein illustrated in Fig 1B of the manuscript shows that PC9/GR1, PC9/DR1 and PC9/OR3 cells (which were further studied) express higher levels of full-length PPP3CB protein (PPP3CB exon 16 isoform) compared to parental PC9 cells. Therefore these results demonstrate that PPP3CB exon 16 splicing correlates with increased PPP3CB protein levels *in vitro*. Because PPP3CB western blots also revealed the presence of additional bands with lower molecular weight (see SourceDataFileForFigure1 provided with

the revised manuscript), IHC analysis was not possible to specifically study PPP3CB Ex16. Therefore, we decided to use a RNA-based technology to study the expression of PPP3CB in human samples.

6. The authors are advised to include the original flow cytometry analysis that was used to create bar diagrams in multiple figures. How the gates have been chosen for the analysis.

Original flow cytometry data including the gates chosen for the analysis are now provided in the source data files associated with each figure in the revised manuscript.

7. Loading controls for the RT-PCR experiments are missing.

GAPDH loading controls have been added for the RT-PCR experiments in the revised Figures 1A, 1C, 2C and Fig S1, and show no difference between samples.

8. The authors should quantify protein levels and provide the statistical analysis for the western blot data in all figures, including supplemental, since some of the conclusions are not so obvious as stated

We have now quantified and performed statistical analyses of the relevant protein variations that support the main conclusions of the paper in all figures, including supplemental. The results are presented in the source data files associated with each figure in the revised manuscript and confirm the conclusions presented in the manuscript.

9. The control for tumor cell viability with the Cyclosporin A alone is missing in Fig 3A

Control for cyclosporin A alone has been added in MTS assays of the revised Fig3A. The results show that Cyclosporin A has moderate toxic effect at the concentrations used in our cellular models.

10 and 11. RNAScope data in human samples are not convincing, and the quality of these images is poor to make any definitive conclusions. It is unclear whether there was a negative control for the background staining. There is no conclusive visual difference in the number of dots between pre- and post-treatment groups in patient's tumor section. High magnification images for pre-treatment samples were not provided

We apologize for the poor quality of the images that is probably due to pdf conversion of the entire manuscript. We provide with the revised manuscript an extra ppt file (called « for reviewer only ») containing the original high magnification images illustrated in Fig 2D as well as additional examples of pre- and post-treatment samples showing increased PPP3CB mRNA staining in post-treatment samples. In the original figures 2B and 2D we do acknowledge that we illustrated only some examples of PPP3CB dots in each image. In the revised Fig 2B and 2D we now illustrate all the detected PPP3CB dots with arrows. The results show a clear visual difference in the number of dots between cells transfected with control or siex16 siRNAs (revised Fig 2B), as well as between pre-treatment and post-treatment samples (revised Fig 2D). As requested, higher magnification image for pretreatment sample has been added in the revised Fig2D.

As reported by the manufacturer, the BaseScope™ v2 assays are based on ACD's patented signal amplification and background suppression technology which explain why a negative control for the background staining is not required.

12. From the clinical point of view, it is not clear how beneficial for cancer patients will be the use of calcineurin inhibitors since those inhibitors are used as the immunosuppressors after organ transplantation.

Indeed, calcineurin inhibitors are well known as immunosuppressants and widely used as anti-rejection treatment in organ transplant setting. However, cyclosporin A used at lower doses has also shown interest as an anti-tumor agent in cancers, including lung cancers, in combination with chemotherapy based on platinum salts (Ross et al. 1997, *Lung Cancer*, 18(2):189). Cyclosporin A could have an immunosuppressive effect on a population of regulatory T lymphocytes, which can play a pro-tumoral role in some cancers (Flores C. et al, 2019, *Frontiers in Immunology*, March 2, 10:588). Consistent with the potential interest of using cyclosporin A as an anti-tumor agent, new clinical trials are underway in triple negative breast cancer where CsA is used alone (NCT06246786), as well as in colon cancer in combination with Selumetinib (a mek inhibitor) (NCT02188246). Our data indicate that such combination with CsA and MEK inhibitor could also be relevant in lung adenocarcinoma patients with acquired resistance to EGFR TKI.

13. The toxicity of the new combination of drugs should be provided for the animal experiments. These important data are completely missing.

The new combination of drugs had no significant impact on body weight (Fig S4C) and we did not observe macroscopic abnormality of the organs when the mice were sacrificed (data not shown). Of note we also tested higher drug concentrations with the same results. Therefore, at this stage of the study, these results reveal absence of acute toxicity of the new combination of drugs.

15. The title of the manuscript does not really reflect the content of the manuscript and should be modified

We thank the reviewer for this remark. We have modified the title of the manuscript as following : « PPP3CB overexpression mediates EGFR-TKI resistance in lung tumors via calcineurin/MEK/ERK signaling”.

Adress Reviewer 3' comments

Comments on Novelty/Model System for Author

The authors used 10 μ M Cyclosporin A (CsA) in their experiments which is higher than the generally accepted range for in vitro studies. Typically, CsA concentrations range from 0.1 μ M to 5 μ M to avoid non-specific toxic effects. The high concentration used in this study could lead to non-specific effects, reducing the specificity and interpretability of the results. It is recommended to reevaluate the experiments using CsA concentrations within the appropriate range (0.1 μ M to 5 μ M) to ensure specific calcineurin inhibition and avoid potential confounding effects due to high drug toxicity. Additionally, the authors used 5 μ M trametinib, which is significantly higher than the commonly used concentrations for similar studies. Typically, trametinib is used at concentrations less than 100nM to specifically inhibit MEK without off-target effects. The high concentration raises concerns about non-specific effects and potential toxicity that could confound the results. Moreover, the authors used 100nM trametinib in Fig. 4, which is within the acceptable range, but this inconsistency in concentrations needs clarification. This variation in drug concentration accross experiments could be perceived as an intentional effort to produce results that fit the desired narrative.

We want to specify here that the CsA and trametinib concentrations indicated in Fig3A and 5A (MTS assays) panels do not apply to all other panels of figures 3 and 5. We apologize if this has led to confusion for the reviewer because the drug concentrations used in apoptosis quantification, clonogenic assay and spheroid culture were different and lower than those used in MTS assays. In fact, in order to evaluate the potential positive effect of CsA/trametinib in association with EGFR TKI, the concentration of CsA/trametinib was adapted for each drug and each type of assay (MTS, apoptosis quantification, clonogenicity, spheroids) in order to have a minimal effect on the inhibition of cell growth when each drug was used as monotherapy. Accordingly, the concentration was different depending on the assay. So, in MTS assays **in Fig 3A**, although CsA concentration was high in PC9/OR cells (20 μ M) to see a biological effect, it was only 1 μ M for all PC9/GR, PC9/DR and PC9/OR cells in both apoptosis quantification (Fig3B) and clonogenic assays (Fig 3C), and 5 μ M in spheroid assays (Fig 3D). **In Fig 5**, although CsA/trametinib concentration was high in MTS assays (10 μ M and 5 μ M respectively, Fig 5A), it was only 0.1 μ M and 0.5nM respectively in clonogenic assays (Fig5B) and 2 μ M and 5nM respectively in spheroid assays (Fig 5C). Hence these last concentrations were within the appropriate range to ensure specific target inhibition without off-target effects. In all these assays, a clear anti-tumor effect was observed when the drugs were combined with osimertinib, in agreement with the MTS data. Currently, we do not really explain why increasing concentration of drugs are required in MTS assays compared to all other drug combination assays. However, as we confirmed the results in multiple assays as well as *in vivo*, we are confident with our conclusions that PPP3CB overexpression contributes to acquired resistance to EGFR TKI via activation of a calcineurin/MEK/ERK pathway. To avoid any confusion for readers, we have decided to include drug concentrations in all panels of the figures, and to explain in the revised manuscript how these concentrations were selected.

Remarks for author

The most important question of this paper is whether PPP3CB is dominantly associated with osimertinib resistance, especially in PC9-OR3. Firstly, the authors should exclude the possibility of known osimertinib resistance mechanisms, such as EGFR secondary mutations, BRAF, KRAS mutations, and bypass signaling activation, such as MET and HER2. Notably, in PC9-OR cells, 10 μ M CsA did not sensitize the cells, whereas 1 μ M CsA was enough to sensitize PC9/GR cells (Fig. 3A). This suggests that PC9-OR3 cells may not be dominantly dependent on the expression of PPP3CB compared with PC9/GR, indicating different resistance mechanisms other than PPP3CB. The author must clarify to what extent PPP3CB is involved in EGFR-TKI resistance and whether it is only involved in apoptosis in some cell populations.

In the materials and methods section as well as in the text of the original manuscript, we mentioned that resistant PC9/OR3 cells do not carry MET and HER2 amplification, T790M and C797S EGFR secondary mutations or KRAS, BRAF, HER2 or PIK3CA mutations, thereby excluding the possibility of these known genetic mechanisms of resistance to osimertinib. We agree that higher concentration of CsA are required to sensitize PC9/OR cells as compared to PC9/GR cells. Interestingly, our results also showed that the increase of intracellular calcium level was higher in PC9/GR cells treated with gefitinib (Fig S3E) as compared to PC9/OR cells treated with osimertinib (Fig 4J). This suggest that calcineurin activation may be more important in gefitinib-treated PC9/GR cells than in osimertinib-treated PC9/OR cells. We think that this could contribute to the difference of sensitivity to CsA treatment as PPP3CB protein level was similar in PC9/GR and PC9/OR cells (see quantification in the SourceDataFileForFigure1 provided with the revised manuscript).

Fig 1C. The authors show the effect of siEx16 on PC9-EGFR-TKI resistant cells in Fig1C and the effect of overexpression of PPP3CB-GFP on PC9 cells, indicating the causal relationship between the resistance and the expression of PPP3CB. How about the effect on cell viability ? It would strengthen the conclusion if you could also show data on how PPP3CB expression impacts cell growth.

Using MTS assays, we now show that the knock-down of calcineurin (siEx16) diminishes the viability of resistant cells following EGFR TKI treatment (revised Fig 1D and see also response to next point). On the other hand and despite several attempts, we were not able to clearly see a difference of viability (MTS assay) between cells overexpressing GFP or PPP3CB GFP and treated with EGFR TKI for 96 hours. Nevertheless, to go further, we have performed long term clonogenic assays in these cells treated with osimertinib for up to 13 days. The results show that PC9-PPP3CB-GFP cells were more resistant to osimertinib-induced cell growth inhibition than PC9-GFP (revised Fig 1F). Altogether, these results further strengthen the role of PPP3CB overexpression in resistance to EGFR-TKI.

Fig 3A The authors show the combination effect of EGFR-TKIs and CsA. In PC9/GR cells, 1 μ M CsA was enough to sensitize the effect of gefitinib : however, in PC9/OR cells, even 10 μ M CsA was not effective, suggesting that calcineurin might not be associated with the resistance mechanisms. Generally, 10 μ M seems too high. If you aim to prove that calcineurin predominantly contributes to the resistance mechanisms, you should knock-down the expression of calcineurin and evaluate the effect of EGFR-TKIs by MTS assay.

As suggested by the reviewer, we have knocked-down calcineurin (by the use of siEx16) and studied the effect of EGFR-TKIs on cell survival by using MTS assay. The results show that the neutralization of calcineurin clearly inhibits the growth of PC9/GR, PC9/DR and PC9/OR cells following EGFR TKI treatment (revised Fig1D). Therefore, these results demonstrate that calcineurin overexpression predominantly contributes to the resistance mechanisms including in cells resistant to osimertinib.

Fig 5A-E. The authors use 5 μ M trametinib. However, this concentration is too high, suggesting that trametinib might suppress targets other than MEK. Generally, the concentration of trametinib should be less than 100nM. Interestingly, you used 100nM in Fig 4. I wonder why you used different concentrations. Have you considered using a lower concentration of trametinib and showing the p-MEK levels when using trametinib by WB ? Additionally, 10 μ M CsA also seems too high. Reevaluating with appropriate concentrations is recommended.

Please see our response at « Comments on Novelty/Model System for Author »

September 19, 2024

RE: Life Science Alliance Manuscript #LSA-2024-02873-TR

Dr. Sylvie Gazzeri
INSERM U1209
Allee des Alpes
site Sante
Grenoble 38000
France

Dear Dr. Gazzeri,

Thank you for submitting your revised manuscript entitled "PPP3CB overexpression mediates EGFR-TKI resistance in lung tumors via calcineurin/MEK/ERK signaling". We would be happy to publish your paper in Life Science Alliance pending final revisions necessary to meet our formatting guidelines.

- please be sure that the authorship listing and order is correct
- please add the Twitter handle of your host institute/organization as well as your own or/and one of the authors in our system
- please incorporate the supplemental materials section into the main Materials & Methods section

Figure Check:

- in figure S2, you have panels A and B in the figure, but these are not referred to in the figure legend. Please correct.
- there are 2 related manuscript files, but it is unclear what these are. One of them looks like it could be uploaded as a Graphical Abstract instead.

A. FINAL FILES:

B. MANUSCRIPT ORGANIZATION AND FORMATTING:

Sincerely,

September 24, 2024

RE: Life Science Alliance Manuscript #LSA-2024-02873-TRR

Dr. Sylvie Gazzeri
INSERM U1209
Allee des Alpes
site Sante
Grenoble 38000
France

Dear Dr. Gazzeri,

Thank you for submitting your Research Article entitled "PPP3CB overexpression mediates EGFR-TKI resistance in lung tumors via calcineurin/MEK/ERK signaling". It is a pleasure to let you know that your manuscript is now accepted for publication in Life Science Alliance. Congratulations on this interesting work.

DISTRIBUTION OF MATERIALS:

Again, congratulations on a very nice paper. I hope you found the review process to be constructive and are pleased with how the manuscript was handled editorially. We look forward to future exciting submissions from your lab.

Sincerely,
